# Time-Restricted Feeding and Weight Loss in Obesity: A Mini Review on the Impacts on Skeletal Muscle Mass

Guilherme Correia Ferri Antonio [1] , Adelino Sanchez Ramos da Silva [2] , Ellen Cristini De Freitas [2] and José Rodrigo Pauli [1,3,*]

[1]  Laboratory of Molecular Biology of Exercise (LaBMEx), University of Campinas (UNICAMP), Limeira 13083-970, Brazil; guilhermecorreia.gcf@gmail.com
[2]  School of Physical Education and Sport of Ribeirão Preto, University of São Paulo (USP), Ribeirão Preto 14040-907, Brazil; adelinosanchez@usp.br (A.S.R.d.S.); ellenfreitas@usp.br (E.C.D.F.)
[3]  Laboratory of Cell Signaling, Obesity and Comorbidities Research Center (OCRC), University of Campinas (UNICAMP), Campinas 13083-970, Brazil
*  Correspondence: rodrigopaulifca@gmail.com

**Abstract:** Across the globe, obesity is a significant concern for public health, a disease characterized by excessive accumulation of body fat, with a negative impact on health. Time-restricted feeding (TRF), in which food accessibility is restricted to a variable period of 8–10 h, especially in the active phase, inducing a prolonged fasting period, is a strategy with potential effects in preventing obesity. Evidence in preclinical studies demonstrated that TRF attenuates the impacts of metabolic disturbances related to high-fat diet feeding in rodents. Through these discoveries, there has been growing interest in revealing the effects associated with TRF in preventing obesity and its comorbidities, as well as investigating its effects in humans. Although TRF is a promising alternative to reduce the impact of obesity, it is necessary to investigate the results on skeletal muscle tissue. Muscle tissue is important for body energy expenditure; however, caloric restriction can negatively impact protein turnover and induce loss of muscle mass, influencing the basal metabolic rate and weight loss. This mini review examined the scientific literature exploring the effects of time-restricted feeding (TRF) on muscle mass. Although still incipient, the findings related to TRF applied to obese individuals highlight the importance of carrying out dietary control, as well as the consumption and fractionation of proteins, to maintain a balanced protein turnover and, consequently, muscle mass.

**Keywords:** obesity; intermittent fasting; skeletal muscle; protein turnover



## 1. Introduction

The prevalence of obesity is a global public health problem with multifactorial etiology, including genetic, physiological, sociocultural, economic, and environmental factors [1]. Obesity is linked to numerous enduring health disorders, such as diabetes, hypertension, non-alcoholic fatty liver disease (NAFLD), cardiovascular disease, and various cancer types, as well as non-metabolic complications, including anxiety and depression [1].

Despite its constraints, addressing the health consequences of conditions associated with obesity often involves promoting weight loss to enhance both individual health and that of the population. However, weight loss achieved through restrictive diets triggers physiological and psychological "homeostatic pressures", contributing to weight regain in over 90% of individuals [1].

In recognition of the limited effectiveness of restrictive dietary approaches with low long-term adherence, increasing emphasis has been placed on intermittent fasting nutritional strategies with time-restricted feeding (TRF). TRF is a fasting strategy with great applicability, showing excellent results for individuals who face difficulties in adhering to caloric restriction and who fear the possible side effects that a pharmacological treatment protocol may present [2–4].

It is established that obesity is correlated with impairments in muscle performance, resulting in an increased risk of developing functional disabilities, encompassing limitations in mobility, strength, postural control, and dynamic balance [5]. A possible negative result of TRF as a strategy to combat obesity could be the inhibition of muscle protein synthesis induced by post-prandial stimulation. As TRF decreases food intake, including protein, for some time, there is a hypothesis that this strategy could lead to a negative nitrogen balance and, consequently, to loss of muscle mass [6,7]. Therefore, this mini review aimed to present scientific findings regarding the impact of TRF on managing and preventing obesity, while emphasizing its potential role in preserving and sustaining skeletal muscle mass. In this way, health professionals who carry out interventions with obese people can create personalized intervention strategies to reduce body weight without damage to skeletal muscle mass.

## 2. Time-Restricted Feeding

Intermittent fasting has been applied as an important tool for treating obesity and related diseases. This feeding protocol comprises several different forms of food intake, such as daily or weekly intake. Among the protocols studied, the three most relevant comprise complete fasting on alternate days (ADF): which involves deprivation of food on alternate days (fasting on days); modified alternating diets (MADF): which adopts different variations of ADF; and finally, time-restricted feeding (TRF): where food intake is restricted to specific times, with fasting for a long period. The scientific literature considers TRF as one of the most widely practiced fasting strategies to improve health and body composition through caloric restriction. TRF can range from 4 to 12 h, but an 8 h feeding period with a daily 16 h fast is typical [8,9].

Unlike most forms of intermittent fasting (IF), TRF does not necessarily have to be performed with a reduced caloric intake, as the potential benefits are linked to the time of day when the meals will be completed. The influence of the circadian system could explain the time-dependent effects observed in TRF interventions. The energy metabolism of glucose and lipids is regulated by the circadian cycle, which has an up and down regulation at different times of the day. For example, in humans, insulin sensitivity and the thermic effect of food exhibit greater prominence in the morning compared to the afternoon or evening. This observation implies that energy metabolism is optimized for food consumption during the daytime [10].

In a study conducted by Keim et al. (1997) involving women following a controlled weight-reduction regimen, it was observed that consuming the main meal in the morning (AM) resulted in more significant weight loss in comparison to finishing it in the evening (PM) [2]. Furthermore, consuming the main meal in the morning was associated with improved preservation of lean body mass. Thus, incorporating PM meals in a weight loss program may play a crucial role in reducing the loss of fat-free mass. However, for meals aligned with the circadian rhythm, increasing morning food intake may be more effective for weight loss [11]. This finding indicates that the efficacy of TRF interventions may rely on distinct mealtime strategies, necessitating different approaches for each specific objective.

## 3. TRF and Obesity

Lifestyle changes are the basis for obesity prevention and treatment strategies. These alterations include decreasing energy intake to approximately 500 kcal, increasing physical activity, and behavioral change techniques. Conventional calorie-restricted diets may have limited long-term success, with adherence levels typically lasting for 1 to 4 months, followed by significant weight regain in approximately one year [12].

As a way of preventing and treating obesity, TRF has been used as a fasting strategy with great applicability, offering a variety of nutritional therapy options for those who face difficulties in adhering to caloric restriction and who are concerned about possible side effects derived from a pharmacological treatment protocol. Preclinical studies provided scientific evidence demonstrating that the implementation of TRF can offer protection to

obese rodents that are fed a high-fat diet. This protection is manifested through mitigated increases in body adiposity and reduced endocrine-metabolic disorders [13–17].

In this context, Hatori et al. (2012) studied whether obesity and metabolic diseases are due to a high-fat diet (HFD) or the interruption of metabolic cycles. For this, mice were fed ad-lib for 8 h TRF in the dark phase of an HFD for 8 h a day. Consequently, the researchers noted that mice following the TRF regimen consumed a similar amount of HFD calories as those with unrestricted access. However, the TRF group exhibited protection against obesity, hyperinsulinemia, fatty liver, and inflammation.

A study by Olsen et al. (2017) reported that animals with obesity induced by a high-fat diet submitted to a TRF regimen presented a restriction in weight gain when compared to free-fed animals, despite the same caloric intake levels. TRF also efficiently prevented excessive weight gain and metabolic diseases in mice without a circadian cycle in a food access protocol restricted to 10 h during the dark phase [10]. Overall, current evidence is positive, as TRF is associated with decreased body weight, improved glycemic homeostasis and insulin signaling, reduced inflammation, and a favorable effect on dyslipidemia [14,17–19]. Studies have also shown that TRF can prevent the development of NAFLD [10,13,20,21].

Following from the success of the effects of TRF when applied to animals, studies were developed using this dietary strategy in humans, which also showed efficient results in obesity. This fact was observed by Cienfuegos et al. (2020), where obese adults on a TRF protocol ranging from 4 to 8 h over 8 weeks successfully reduced weight, insulin resistance, and oxidative stress versus controls [22]. Positive data were also verified in a study by Jamshed et al. (2022), in obese individuals, where an 8 h food window was more effective for weight loss when compared to a group that performed a conventional food restriction protocol for 12 weeks [8].

In a study by He et al. (2022), the effects of a low-carbohydrate diet (LCD), an 8 h TRF protocol, and their combination were investigated over three months on body weight and abdominal fat area. The LCD treatment and the 8 h TRF protocol significantly reduced body weight and the subcutaneous fat area. However, only the TRF and the combined intervention reduced the visceral fat area (VFA), fasting glucose levels, uric acid (UA) levels, and improvements in dyslipidemia. In conclusion, an 8 h TRF can be regarded as an effective therapeutic approach for addressing metabolic syndrome [23].

Even with the benefits mentioned above, there are concerns regarding TRF and muscle mass homeostasis. A deficiency in caloric and protein intake can have a catabolic effect, in which degradation is more significant than protein synthesis, inducing loss of muscle mass. A possible negative result of TRF could be the inhibition of muscle protein synthesis induced by post-prandial stimulation.

## 4. TRF, Obesity, and Skeletal Muscle

Skeletal muscle (SM) constitutes 40 to 50% of body mass and is our most abundant tissue. SM is essential in contractile force production, post-meal glucose uptake, and resting energy expenditure. Increased adiposity from obesity is commonly associated with general muscle dysfunction. Furthermore, obesity is closely related to detrimental impacts on muscle metabolism, maintenance of insulin sensitivity, regulation of ectopic fat deposition, and regulation of basal metabolic rate [5].

Skeletal muscle balance is determined by protein turnover. This phenomenon is due to the constant ratio of protein synthesis and breakdown balance impacting SM mass. Nutritional status stands out among the factors that significantly influence protein turnover. A deficiency in caloric and protein intake can have a catabolic effect in which degradation is more significant than protein synthesis, inducing loss of muscle mass.

Considering these factors, nutritional approaches employed for the prevention and management of obesity must, above all, consider the potential effect on protein turnover, preserving the integrity of muscle mass, and ensuring an adequate intake of calories and proteins to maintain homeostasis and inhibit skeletal muscle catabolism.

A possible negative result of TRF as a strategy to combat obesity could be the inhibition of muscle protein synthesis induced by post-prandial stimulation. As TRF restricts food intake (including protein) for some time, there is a hypothesis that this strategy could lead to a negative nitrogen balance and, consequently, to the loss of lean body mass.

A study by Chow et al. (2020) showed important results on this phenomenon. The study aimed to identify, in adults [(17 women/3 men) ~45.5 years, body mass index 34.1 kg/m$^2$ (7.5)], whether a TRF protocol with a food window of 8 h vs. non-TRF for 12 weeks would be more effective in facilitating weight loss, changing body composition, and improving metabolic measures. Among the reported results obtained through dual X-ray absorptiometry (DXA), the authors observed greater loss of total body weight, visceral fat, and lean mass in the non-TRF group compared to the TRF group. The loss of lean mass in the TRF group was in the legs [−3.7% (3.6)], which was significant compared to pre-intervention and non-TRF. TRF did not result in considerable trunk or arm lean mass loss. When comparing the pre- and post-intervention metabolic measures, it was also observed that TRF had significant results on fasting glucose [−7.7% (6.9)] and fasting triglyceride concentration [−23.6% (21.7)] (both $p < 0.05$). The TRF intervention did not change HbA1c or insulin sensitivity relative to the pre-intervention or non-TRF group. Despite these results, no significant differences were observed between the TRF and non-TRF groups concerning metabolic measurements.

The above reported data allow the observation of the interference of TRF as a means of preventing and treating obesity, in the integrity of lean body mass. As previously mentioned, an important point to consider is the control of caloric and protein intake to preserve muscle mass. In this case, the TRF protocol did not adopt a calorie-restricted nutritional strategy, and diet intake, diet quality, and energy intake instructions were not provided. In addition, the only monitoring was performed through food intake documented on an application during the protocol. This fact may have contributed to the loss of lean mass, since studies show that fractioning and the regular intake of protein in healthy individuals (4 × 20 g every 3 h) are more relevant to maximize daily rates of muscle protein synthesis and to maintain muscle homeostasis compared with smaller amounts of protein several times a day (8 × 10 g) or two high-protein meals (40 g) [24,25].

A TRF protocol with control of caloric and protein intake was recently reported in a study by Parr et al. [7]. The study aimed to identify whether an 8 h isoenergetic and isonitrogenous TRF as an obesity treatment would reduce the myofibrillar protein synthesis rate compared to 12 h feeding, leading to reduced lean body mass in overweight men. Eighteen individuals were selected, middle-aged men (35 to 55 years) with overweight or obesity (BMI: 25–35 kg/m$^2$), who were randomized into two groups, namely, the TRF group (10:00 to 18:00) and the control group (CON; 8:00 a.m. to 8:00 p.m.) for 10 days. The total energy expenditure of the individuals was calculated to distribute macronutrients according to energy need (56% carbohydrates, 30% fat, and 14% protein 1.0 g/kg/d) [7].

For the TRF group, meals were divided into three periods: breakfast at 10 am, lunch at 2 pm, and dinner at 6 pm, following a suggested 8 h food window. In addition to age, the initial measurements of body composition, physical activity, continuous glucose monitoring, fasting blood samples, and blood pressure were not different between individuals in the CON and TRF groups. The authors also observed that the fractional synthesis rate of myofibrillar protein evaluated in saliva and plasma was similar for both groups after the 10-day experimental period.

In addition, whole body dual energy X-ray absorptiometry (DXA) analyses showed a significant decrease in total body mass, fat mass, and lean mass during the testing period, with a more significant decrement in the body fat percentage and total fat mass in the CON group when compared to TRF, while the decrease in lean mass was greater for TRF individuals, with this reduction being greater in the trunk region. Changes in the total 24 h area under the curve (AUC), measured by a continuous glucose monitor for fasting glucose, as well as the mean and standard deviation of 24 h glucose concentrations, were more pronounced in TRF than in CON. It is worth noting that the 2 h delay in breakfast for TRF

decreased the pre-meal glucose concentration. The 2 h advance of dinner decreased the peak post-prandial glucose concentration, leading to a decrease in the AUC 2 h increment and peak.

In contrast to their hypothesis, the authors concluded that food consumption within a restricted 8 h window did not impair MyoPS rates and improved glycemic control by decreasing post-prandial 24 h glycemic excursions. However, even with controlling caloric and protein intake, the results showed a statistical reduction in muscle mass in the TRF group compared to the control group. This finding may be related to the amount of protein used in the protocol (1 g/kg of body weight), since some studies showed interesting results in preserving fat-free mass and decreasing fat mass in obese adults when slightly higher amounts of protein were offered (~1.2 g/kg body weight) [12,26].

In addition, both studies adopted a short-term protocol with a small number of participants, which prevents the detection of solid results [6,7]. As previously mentioned, dietary protocols with slightly higher amounts of protein (~1.2 g/kg of body weight) for obese patients may be interesting for maintaining muscle mass in individuals undergoing a TRF protocol [12,26]. Therefore, future papers should focus on ensuring that, during the dietary window, patients on TRF ingest the minimum amounts, or even slightly higher amounts of protein than recommendations, to promote the maintenance of skeletal muscle mass. Another important point to note is that there was no evaluation of muscle function in the analyzed studies, in which only the loss of muscle mass was evaluated via DXA, and it is also interesting to consider this analysis in future studies.

Furthermore, Oxfeldt et al. (2023) evaluated the impact of 10 days of low energy availability (LEA) on myofibrillar and sarcoplasmic muscle protein synthesis in young, trained women. The results indicated that LEA impairs the structure of myofibrillar and sarcoplasmic muscle proteins in this population. These results suggest that LEA may have negative adaptations for skeletal muscle, limiting gains or the ability to repair muscle, highlighting the importance of ensuring adequate energy availability in female athletes [27]. These findings demonstrate the importance of investigations involving obese individuals who adopt TRF and physical exercise, in order to investigate the impacts on muscle mass. Figure 1, in a summarized manner, demonstrates the potential effect of TRF in contributing to weight loss; however, it also highlights the possible impact of this intervention on the reduction in muscle mass in the condition of obesity.

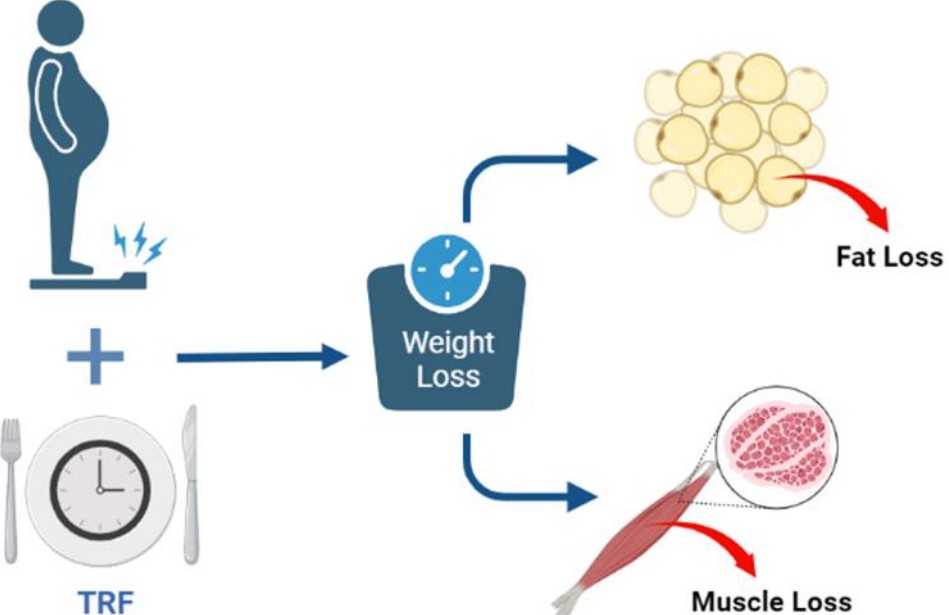

**Figure 1.** Summary of the effects of TRF as a treatment strategy for obesity and its impact on muscle mass. Created with BioRender.com. License number: TU25OBYYYI.

## 5. Future Perspectives

In summary, the studies highlighted that TRF for the treatment of obesity presents benefits for body mass loss and glycemic control; however, a decline in free fat mass can also be observed [6,7]. The main factors that seem to interfere with muscle mass integrity in the TRF protocol are the control of caloric and protein intake. Therefore, dietary control, as well as protein consumption and fractionation, should be carefully controlled to maintain protein turnover and the muscle mass homeostasis should be evaluated in obese individuals submitted to this dietary protocol. It is necessary to emphasize that studies aiming to investigate the effect of TRF on weight loss and that also carried out the analysis of muscle mass in obese people are still scarce and with a very small number of participants. The current mini review aimed to arouse the interest of other researchers in this area, which is of fundamental importance for weight-loss programs. In addition, the findings highlight the importance of complementing the analysis related to muscle mass loss with function tests, allowing a broader understanding of the impacts of TRF on skeletal muscle. Finally, future studies should investigate whether muscle mass loss occurs when the TRF protocol is applied to females. In addition, efforts should be directed to new studies with older individuals with overweight/obesity, where protein catabolism is more likely.

**Author Contributions:** G.C.F.A.—writing—original draft preparation; A.S.R.d.S.—writing—review and editing; E.C.D.F. writing—review and editing; J.R.P.—conceptualization, writing—review and editing, supervision. All authors have read and agreed to the published version of the manuscript.

**Funding:** The present work received financial support from the National Council for Scientific and Technological Development (CNPq; process number 308999/2022-3; 303766/2022-0; 311939/2020-1), and the Coordination for the Improvement of Higher Education Personnel (CAPES; finance code 001).

**Institutional Review Board Statement:** Not Applicable.

**Informed Consent Statement:** Not Applicable.

**Data Availability Statement:** Not applicable.

**Conflicts of Interest:** The authors declare no conflict of interest.

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
