# Peer review of "Time-Restricted Feeding and Weight Loss in Obesity: A Mini Review on the Impacts on Skeletal Muscle Mass"

_2673-4168, doi:10.3390/obesities3030018_

Round 1
Reviewer 1 Report
The topic proposed by the authors, a priori, is interesting. However, the search for this items is scarce. The studies are of short duration, with a small number of patients, practically exploratory; so, the conclusions derived from this analysis may be biased.
In addition, the body composition evaluation systems are not well indicated in the text and the values obtained, beyond the fat-free mass, are not really indicative of loss of muscle mass and function.
Overall, there is not sufficiently homogeneous material to carry out what would be considered a systematic review; therefore drawing risky conclusions at this moment.
Reviewer 2 Report
The presented manuscript is interesting and I have no objections to this review. In my opinion, the main subject of manuscript is discussed in relatively detail. However, I lacked indication regarding the composition of the diet, which contributes the least to the loss of muscle mass during use TRF. The Authors should added a few indication. There is also no translation of the abbreviation IF (p 2, line 72).
Best regards
Round 2
Reviewer 1 Report
The changes made by the authors show sufficient explanations to describe the paucity of robust results to suggest poor protein commitment with TRF diets.
I consider it suitable to be published in this journal.